# Perspectives of Nursing Students on Promoting Reflection in the Clinical Setting: A Qualitative Study

Yasir Alsalamah [1], Bander Albagawi [2], Lisa Babkair [3], Fahed Alsalamah [4], Mohammad S. Itani [5], Ahmad Tassi [5] and Mirna Fawaz [5,*]

1    Emergency Department, Al-Amal Psychiatric Hospital, Qassim, Buraydah 52326, Saudi Arabia; ysalsalamah@moh.gov.sa
2    Medical Surgical Nursing Department, College of Nursing, University of Hail, Hail 55476, Saudi Arabia; b.albagawi@uoh.edu.sa
3    College of Nursing, King AbdulAziz University, Jeddah 22252, Saudi Arabia; lbabkair@kau.edu.sa
4    Department of Nursing, Qassim University, BSN, RN, Riyadh 52571, Saudi Arabia; f.alsalamah@qu.edu.sa
5    Faculty of Health Sciences, Beirut Arab University, Beirut 11-5020, Lebanon; mohammad.itani@bau.edu.lb (M.S.I.); a.tassi@bau.edu.lb (A.T.)
*    Correspondence: mirna.fawaz@bau.edu.lb; Tel.: +96-103-785-199

**Abstract:** Background: Reflection increases meta-cognitive capacities, promotes student-instructor relationships, overcomes the theory-practice gap, and enriches learning. This study aims at exploring nursing students' perspectives on the facilitators of reflective practices in the clinical setting. Methods: Semi-structured interviews were conducted among 21 Saudi nursing students at one major university in Saudi Arabia. Results: Two major themes were prevalent upon thematic analysis, namely, "Personal Drivers of Reflection" which consisted of "Motivation to Learn", "Desire to develop", "Ethical Regard", and "Responsibility towards patients" and "External Drivers of Reflection" which consisted of "Patient characteristics", "Case complexity", and "Competent Instructors". Conclusion: As per the results of the study, nursing students perceived that they underwent reflection as a response to personal motivational and external educational aspects. According to the findings, instructors should assist students through clinical reflection, with a special focus on their interactions and motivation.

**Keywords:** reflection; nursing students; qualitative; perspectives; clinical setting

## 1. Introduction

Nowadays, nurses work in a system that is extremely complicated, active, and ever-changing while providing a diverse array of healthcare functions. Nursing students must enhance their clinical education in order to manage these developments [1]. The foundation of learning through experiencing is reflection, which involves the combination of both knowledge and practice. Many instructors suggested that in order to cope with the present problems in the clinical context, educators should prepare students for reflection on clinical encounters [2]. Reflection has been more prominent in educational facilities over the past years, and it is now recognized as a component in gaining competency. It is a useful and indispensable method for improving professional nursing, and it is broadly utilized in learning and teaching methods in both lecture halls and clinical contexts [3].

Reflection is more than being contemplative; it relates to a mechanism that may help people learn from their experiences. This method entails a critical examination of prior experiences in order to induce change in behavior and improve personal and professional capabilities in the future [4]. Generally, students may scrutinize their behaviors and activities within the experiential context in which they engage through reflection, therefore, resolving the inconsistencies in their performance [5]. Reflection on clinical skills by nursing students during the clinical learning process is critical to the development of such student competencies [6]. Nonetheless, the required abilities for reflection are vague, and

learning reflection is difficult. Students also find the procedure of reflection difficult and feel that it does not occur spontaneously, therefore, it needs a safe and secure atmosphere in which they may cultivate reflection with the assistance of professionals. As a result, facilitation of reflection and reflective capacities is seen as an essential part of professional growth [7]. Students' abilities and capacities increase when they are able to reflect on their everyday occurrences, which can be accomplished by immersing students in an exercise that leverages their consciousness and allowing them to communicate their views [8]. Medical trainers must encourage reflection by assessing students' reflective work and identifying educational obstacles in medical curricula while utilizing handwritten or web-based portfolios. It's crucial to establish a secure setting in which people may reflect on their experiences [9]. Reflection in medical education can be implemented in multitude of ways, depending on the restrictions of the coursework. Many teaching programs require students to keep a reflective journal, and the writings are frequently used for evaluation, thus highlighting the student's weaknesses and encouraging students to improve [10].

As per several pieces of research, reflection improves student-instructor interactions, bridges the theory-practice gap, deepens learning, and strengthens meta-cognitive abilities [11–13]. Regardless of the premise that reflection has been employed in nursing academia for many years and recognized as an academic requirement in the nursing profession that improves students' knowledge and awareness, this subject still requires further investigation, especially in Saudi Arabia, where despite the advances in medical education and research, this topic is still in its infancy. In past years, a considerable discrepancy across instructional experiences and practical nursing services has resulted in a nursing care discrepancy, as well as critique of nurses' care quality in some Saudi hospitals [14]. In order to offer a better understanding of the significance of reflection for implementing more successful programs, it is important to evaluate nursing students' perceptions of reflection. The outcomes of the study can be used to develop a clinical education program that bridges the gap between theory and practice. In Saudi Arabia, the only study that tackled the reflection highlighted the perspectives of preceptors rather than students. The current study was undertaken to explore the nursing students' perspectives on the facilitators of reflective practices in the clinical setting.

## 2. Materials and Methods

### 2.1. Research Design

This study employed a phenomenological explorative methodology in order to explore the nursing students' perspectives on the facilitators of reflective practices in the clinical setting. The methodology used in this research is predicated on Colaizzi's phenomenological framework, which uses respondents' viewpoints and observations to explain the phenomenon in question, culminating in the recognition of common attributes within the sample group rather than personal qualities [15]. The paper was designed and written according to the Consolidated Criteria for Reporting Qualitative Research (COREQ) reporting guidelines [16].

### 2.2. Setting and Sample

A purposive sample of 21 nursing students from the second semester and above from one nursing school in Saudi Arabia and were employed into this study [17]. Students were chosen starting from the second semester because nursing students must complete around 39 credits in the clinical area, which commence in the second semester, and they are regularly confronted with a variety of challenging situations from the outset of their training program, that might drive reflection. In order to acquire profound and comprehensive data, it was sought to choose individuals with the most variety (in terms of semester, average level, and family status). Students who were registered in a bachelor nursing degree at the targeted university, had at least one semester of clinical training program, and were prepared to partake and discuss their insights met the study's inclusion criteria. First-semester students were excluded because they lacked clinical experience. The students

included were enrolled in the full-time study program, and have been practicing reflection throughout their clinical training program through fulfilling a portfolio containing qualitative reflection forms for each clinical task.

The study sample was made up of 21 male nursing students from one nursing school at a major university at Saudi Arabia, where 3 (14.28%) where in the second semester of their first year, 4 (19.04%) where in the first semester of their second year, 5 (23.80%) from the second semester of the second year and another 5 (23.80%) from the first semester of their third year. Only 4 (19.04%) were in the second semester of the third year. The mean age of the students was 21.65 ± 1.27 years (Table 1).

**Table 1.** Student Characteristics.

| Variable | Category | *n* | % |
|---|---|---|---|
| Gender | Male | 21 | 100 |
| Semester/Level | Second semester (1st year) | 3 | 14.28 |
| | First semester (2nd year) | 4 | 19.04 |
| | Second semester (2nd year) | 5 | 23.80 |
| | First semester (3rd year) | 5 | 23.80 |
| | Second semester (3rd year) | 4 | 19.04 |
| Age (M ± SD) | 21.65 ± 1.27 | | |

*2.3. Recruitment and Data Collection*

As researchers were granted access to a registry of university webmail, students were contacted by email. Students were emailed a generic invitation to participate in the research as well as an explanation of the study objectives, and if they were willing to participate, they were asked to return a signed informed consent form. Following introduction, the researchers asked study subjects to participate in hour-long semi-structured interviews using Microsoft teams. The students were approached by the researchers who were not affiliated with their university, and did not teach them any classes. Therefore, the researchers who established contact with the students had no influence on them or the participation process and deciding against it, did not affect their assessments.

*2.4. Interviews*

The interviews were conducted by three of the researchers who were PhD holders and lecturing professors. The researchers who interviewed the students were two males and one female, where all three of them had experience in conducting qualitative research. The researchers had no prior relationship with the students; however, they were introduced upon receiving the email of participation through mutual colleagues and professors who taught the students. The interviews were carried out using virtual meeting technologies by the investigators, who were able to conduct interviews until data saturation was reached. A suitable schedule for the interviews was agreed upon with each participant, especially considering their hectic academic schedules, in order for them to be approachable and provide accurate descriptions of their encounters. The researchers alternated conducting the interviews to avoid the potential of a moderator's domination; Each interview took between 20- and 25-min. Krueger and Casey (2015)'s interview guide (Table 2) was used in order to conduct the interviews as follows:

**Table 2.** Interview Questions.

| Number | Step |
|---|---|
| Introduction Question | What does the term reflection mean to you? |
| Transition Question | What are your impressions on the importance of reflection in clinical practice? |

**Table 2.** *Cont.*

| Number | Step |
|---|---|
| Key Questions | Kindly elaborate on one of your clinical encounters that has prompted you to ponder and reflect more? |
| | Which circumstance led you to reflect more? |
| | What causes you to reflect on these scenarios? |
| | What are the factors that facilitate reflection? |
| | What are the factors that hinder reflection? |
| Final prompt | Do you have anything further to say? |
| Probing Questions | Could you give us a better description? |
| | Could you provide us with a better description? |

### 2.5. Data Analysis

The components of each interview were transcribed and typed shortly thereafter. Each transcription was compared against recordings to ensure authenticity. To gain a thorough grasp of the data, the researchers reviewed the transcripts and played each recording multiple times, extracting relevant data, coding repetitive data, and finally summarizing the data as themes, according to the Colaizzi method. It is worth noting that there are seven steps to the Colaizzi techniques as described in Table 3:

**Table 3.** Colaizzi's Seven Step Method.

| Number | Step |
|---|---|
| 1 | Read the entire interview data |
| 2 | Extract substantial statements |
| 3 | Develop concepts |
| 4 | Assemble the group of themes |
| 5 | Integrate themes into a thorough description |
| 6 | Determine the underlying basis of the phenomenon |
| 7 | Contact respondents for additional information |

Since the interviews were conducted in Arabic, the transcriptions were sent to two official translators that are experts in the field where translation and back translation were carried out and then translations were sent to an external expert to check their authenticity. The transcriptions were anonymized and coded where each participant was assigned pseudonym from S1 to S21. The transcriptions were placed in a sealed envelope, delivered to the translation professionals and were received back in a sealed envelope. The data was analyzed using a standard qualitative content analysis technique. The sections of the interviews that dealt with the participants' encounters with the reflection-facilitators were selected. Three researchers completed their own assessment, after which the researchers convened and discussed their results until they reached an agreement on the emerging themes, all while avoiding bringing their own opinions into the discussion. The quotations were provided in a narrative and insightful terms that captured the spirit of the data provided, and then those phrases were grouped, rearranged, and compiled into qualitative themes, which the investigators considered to guarantee a precise and detailed comprehension of the students' experiences.

### 2.6. Trustworthiness and Credibility

The researchers used several methods in accordance with past research in the field of qualitative analysis in attempt to optimize the study report's credibility and prevent biases from emerging [18]. The researcher's protracted engagement with the data and good communication with the respondents have developed the credibility of the findings. As a result, the investigators worked on the study for over three months. The members' checking was used to compare the consistency of concepts obtained from data and respondents views. An external reviewer with experience in qualitative studies evaluated conformability.

The retrieved themes were given to peers for external review, and their suitability was checked and confirmed, and agreement was reached. Furthermore, highest variation sampling helped in data transferability and authenticity. To ensure credibility, the researcher documented and published the whole study procedure, allowing others to do follow-up investigation. Concurrent data analysis allowed the generation of meaningful hypotheses through concurrent interviews, thus enabling the production of a complete understanding of the occurrences. All of the researchers utilized the same questioning frameworks, posed identical inquiries, and extensively explored any new concepts in order to eliminate blind spots in the findings. Several quotes were utilized to describe the research results, providing the research subjects a real voice [19].

## 3. Results

The phenomenological data analysis has given rise to the following themes and subthemes as stated in Table 4:

**Table 4.** Themes and subthemes.

| Theme | Subtheme |
| --- | --- |
| Personal drivers of reflection | Motivation to learn |
| | Responsibility towards patients |
| | Desire to develop |
| | Ethical regard |
| External drivers of reflection | Case complexity |
| | Patient characteristics |
| | Competent instructors |

### 3.1. Personal Drivers of Reflection

The first theme that emerged from analyzing the verbatim of the participating students pointed towards intrinsic drivers for reflection, where students have felt something within them that have triggered them to go back and think more deeply about their clinical experience and learn from it. This theme had four main subthemes; "Motivation to learn", "Responsibility towards patients", "Desire to develop", and "Ethical Regard".

### 3.1.1. Motivation to Learn

The majority of the respondents indicated they reflected on clinical encounters from a specific position, such as a lack of information or a necessity they were unaware of at the moment. One of the most important aspects in the clinical practice reflection process was students' willingness to learn more about clinical care. When confronted with unfamiliar circumstances, learners started to ponder on clinical experiences and seek knowledge from various resources in order to minimize information gaps. Many of the respondents wished to improve their own expertise in order to give better treatment. Participants' willingness to learn more and thus reflect on their experiences was certainly strong. For instance, one of the students said, " … at many instances in my clinical rotation I feel that I lack deep knowledge about clinical decision making and expertise … this pushes me to make the best out of each clinical encounter by stopping and thinking more about what has been done for this patient or that patient so that I can add that into my clinical experience … " (S18). Another student also proclaimed, " … there's still a lot for me to learn so that I can become a proficient nurse and every time I feel like I lack knowledge in a certain aspect of care it only makes me more encouraged to dig deeper and reflect on my practice skills so that I can provide higher care quality for patients … " (S3). A similar experience was also shared by another student, " … I mean with clinical training you get to detect where exactly is your knowledge and skills are lacking and this should make you feel that you need to fill these gaps … this can only be done by reflecting on where did you go wrong or where you could have done better or thinking about which aspects of patient care you still

do not know much about . . . this motivation to learn more about patients and clinical care makes me want to reflect every piece of new knowledge on my practices . . . " (S21).

### 3.1.2. Responsibility towards Patients

A substantial number of students have expressed that their sense of responsibility towards providing patients with their needs and answering their inquiries is a major driver for reflection. They have also indicated that they do not want to be a kind of nurse that is just performing technical procedures. This sense has led students to reflect on their clinical experiences and knowledge as they want to be completely responsible for their patients and doing them good. For example, one of the students said, " . . . my patient is my responsibility even though I am still a student but I need to learn how to take responsibility for my actions with the patients . . . so I need to reflect on my practices, skills and knowledge so that I can learn more and be read for more complex cases and provide the patient with their needs . . . " (S5). Another student also shared, " . . . I don't want to be unable to take the proper actions when the patients need me or when they need answers on something . . . I feel as a student nurse responsible just like a graduate registered nurse for the patients' condition . . . that what makes me want to think more on my level of performance . . . " (S12).

### 3.1.3. Desire to Develop

Another subtheme that was prevalent as an internal driver for reflection on clinical experience revolves around the students' strong desire to develop professionally into highly competent registered nurses. The students expressed their urge to enhance their critical thinking, decision making and practical skills so that they evolve into highly competitive nurses in the labor market. One of the students said for instance, " . . . I reflect on my clinical experiences because when I graduate I want to be not just any other nurse . . . I want to be an outstanding example of the highly professional nurse who continually works on developing themselves through increasing their knowledge and skills by reflecting on the areas needing improvement and thus develops into a higher standard of care . . . " (S9). Another student also indicated, " . . . I aspire to be an international level nurse . . . a nurse that can practice in any type of challenging and complex setting . . . a nurse that can be part of the leadership team . . . ; therefore, to be that, I need to always reflect on my performance to prove myself to be more competent than before and work on my areas of weakness . . . " (S20).

### 3.1.4. Ethical Regard

Moreover, the students who took part in this study have expressed the strong ethical and moral value of their work and the need to be up to the expectations of their personal, professional, cultural, and religious values in their work. The majority of the students felt that their values have driven them to reflect on what they have practiced, how they dealt with the patient, the family, and their colleagues so that they can detect any wrongdoing and make sure to avoid such encounters in the future. For instance, one of the students said " . . . my work as a nurse is guided not only by professional ethics but also by moral and cultural values too, which makes me in need to always relate my practice and performance to, so that I can be sure that I am doing the right thing . . . I mean I would want to be treated in a dignified manner if I was ever in the place of the patient . . . " (S1). In addition, another student shared a similar anecdote, " . . . God is watching everything we do and my religion tells me that I need to do the most good to the people around me . . . that's why I need to always keep myself in check and make sure that my performance is up to the level that will make the patient better and not do harm, whether clinically or even psychologically . . . " (S15).

*3.2. External Drivers of Reflection*

The other main theme that was prevalent among the students who took part in this study related to the extrinsic drivers of reflection. These external motivational factors for reflection revolve around the clinical environment in which the students are practicing which included patients, clinical preceptors and the healthcare team with which the students are in constant contact.

3.2.1. Case Complexity

The first extrinsic driver that was prevalent among the nursing students who took part in this study showed that the students were motivated by the complex cases rather than the common ones to reflect on their knowledge and skills. The students shared that the more the case challenged them, the further they felt that they need to reflect on how they performed and see what they would do differently, especially regarding carrying out related research. For instance, one of the students proclaimed, " . . . new difficult cases make you want to know more about the management and when you handle such cases it makes you feel that you need to always check how did you do and go back and think about all the steps you took and it motivates you to become more proficient in handling such situations . . . " (S4). Another student also said, " . . . it is always very interesting to handle the complicated and heavy cases as you feel the sense of responsibility and achievement that makes you want to reflect on your performance and make it better . . . this proves to me the worth of the work we are doing . . . " (S19).

3.2.2. Patient Characteristics

Not only the details of the disease itself has motivated the participating students to reflect on their clinical practice but rather the patients themselves and their sociodemographic characteristics and interaction patterns have also played a key role in driving and facilitating students to reflect more on their performance. Such characteristics include patient's age and educational level. For example, one of the students shared, " . . . one time I had a patient who is 19 years old . . . this made me put myself in their shoes and pay attention to more details regarding the case and the management as his case was truly critical and I was thinking I can be in his place at any moment in life . . . this make me think a lot about my work and the value of care . . . I felt I want to read more about it afterwards and I did . . . " (S2). Another student also had a similar experience, " . . . one time I was assigned to care for a child who had a very serious condition . . . it made me think a lot about the importance of each step of care I was giving . . . I have never seen such a case at such a small age . . . when I went back home and had to prepare a care plan assignment about her case, I found myself thinking a lot about everything we did for her . . . " (S10). Another student has addressed the knowledge of the patient saying, " . . . one time I had a patient who has been chronically ill and also works in the healthcare field . . . that moment made me feel that I had to be very knowledgeable to be able to care for this patient as I can't have less information about the patient case than the patient themselves . . . this had me doing a lot of research every time I handle a case . . . " (S14).

3.2.3. Competent Instructors

Another external driver for reflection was prevalent in the experiences expressed by the students, where they have indicated that the instructional methods, communication approaches, and leadership style that have been adopted by the clinical instructors that were mentoring them at the clinical site are highly influential factor, that encouraged them to reflect on their performance. One of the anecdotes was, " . . . our instructor is quite an inspiration . . . he sets an excellent example for the competent and highly qualified registered nurse . . . this makes me want to be like him when I graduate . . . and it encourages me to reflect on my clinical practice so that I can develop into a professional nurse like him . . . " (S7). Another student also shared a similar testimonial, " . . . the clinical instructor motivates us to be not only skillful but also knowledgeable and have critical

thinking skills to integrate theory and practice . . . and he demonstrates this through his work with us and with his patients and colleagues . . . this alone makes me think that I want to be this skillful and professional . . . thus I reflect on my practice skills and knowledge I use in caring for my patients so that I can advance myself . . . " (S16).

## 4. Discussion

The outcomes of a research in which student nurses in Saudi Arabia were interviewed through in-depth qualitative approaches are presented in this paper. According to the findings, the drivers of reflection ranged from personal and extrinsic levels. Motivational parameters were reported by the students to be key in promoting their reflection on clinical skills on an individual basis. This study showed that the students think that reflecting on clinical training progress and development is facilitated by their desire to obtain practical knowledge, responsibility to patients, and students' inclination to be influenced by their values. This is consistent with Reljić et al. (2019) who have found that students carrying out reflective practices have expressed to have higher level of knowledge, critical thinking abilities, advanced practices and relationship with patients; outcomes which have driven their reflection [20]. Our results are also in line with Choperena et al. (2019) who have also indicated the value of reflective practice among student nurses in enhancing their knowledge, skills, and intellectual abilities [21]. In addition, Artioli et al. (2021) has published results that corroborate our findings, where this study has explored qualitatively the perspectives of students on the use of reflective practice and has shown that students were driven towards reflection to enhance their knowledge and practice capabilities in order to become more competent nurses and develop professionally [22]. Contreras et al. (2020) also discovered that self-reflection combined with clinical knowledge and expertise reduces anxiety, allowing nurses to build competence in real-world settings [6]. As per the research carried out by Hwang et al. (2018) on nursing staff, there was a link between reflection, intellectual growth, and career development; additionally, it was discovered that reflection improves mental wellbeing. Nursing students receive a sense of efficacy and value when they identify a reflection occasion and succeed in their job in a stressful clinical learning setting. Students attempt to obtain information that may be applied in a variety of clinical contexts by reflecting on clinical training [23]. Consequently, if Dewey once defined learning as a pathway that consists of a mixture of encounters and reflection, one should now define learning as a procedure that consists of a mixture of experience, reflective thinking on encounter, and application of the findings in other scenarios [24]. Thus, the findings of the study highlight the importance of emphasizing the significance of professional values such as personal responsibility and intrinsic motivational factors. Successful reflection is difficult without factors such as practice autonomy, suitable supporting framework, adequate evaluation, activity significance, and reasonable challenging scenarios [25]. According to the results of the data analysis, a qualified trainer might help students in clinical practice with their reflection process. Trainers can help students reflect on their encounters by allowing them to communicate their recollections and perspectives, conducting dialogues, and directing class discussion. According to Lovell (2018), encouraging activity analysis should be accomplished in conjunction with other determinants to enable reflection in the medical field, such as making sure that trainers are sufficiently educated are able to offer a consistent atmosphere for students, and have an effective relationship with their students. The trainer is responsible for creating such conditions [26]. According to another research, students require the assistance of the trainer in identifying difficulties that require reflection. During the reflection phase, they also require their trainer's advice, direction, and supervision. Upon students' clinical training, instructor's responsibility is critical, and that the instructor should pay attention to the students' discourse, critique their rationalization and perceptions, assess their effectiveness in patient interaction from various angles, and provide prompt and appropriate input. Time expended, attitudes used, responsiveness to patient's talk, analysis of logical path of discussion, and technical abilities are all part of this assessment. As an education provider, a trainer should push students to gain

more from other sources and patients who are similar to them [27]. Allowing students to discuss their results while also reflecting on the reasons, encourages them to speak openly about their flaws, limitations, uncertainty, and confusion, rather than hiding their possible negligence. In this study, students stated that novel situations during the training session give a framework for reflection. Many scholars think that providing a sufficient degree of complicated and demanding scenarios as part of a student's learning process promotes the formation of reflection and cognitive and metacognitive abilities [28]. One of the most essential elements of reflection practice, according to Lawrence et al. (2018) is exciting, unexpected, and puzzling clinical experiences. When confronted with difficult conditions, there is a lack of contact and the creation of a sense of need [29]. Other investigators have noted that reflection occurs as a consequence of being conscious of a need and the normal complexity of that need. This problem arises in the face of complicated and unique situations that are difficult to resolve [30].

*Limitations*

There were several limitations to this research that should be noted. One limitation of the current study was that it only looked at nursing students' perspectives; thus, additional research should look at additional subtopics, such as nursing educators' perceptions of encouraging reflection in clinical settings. Another limitation was that despite students were from various levels of the nursing program they all used the same type of reflection practice, thus there was no insight into different types of reflection provided, and the accounts of the students in first and third year about the same level of reflection might have weakened the data.

## 5. Conclusions

Nursing students in this study have identified both personal and external drivers to practice reflection. Instructors have been perceived by the students to have an important role in guiding students through reflection on clinical experiences, with a particular focus on their relationships and motivating elements. The students also perceived that cooperation amongst instructors, clinicians, nursing staff, and nursing students may create a favorable environment for students to reflect on their encounters through proper communication, as well as a desirable supporting environment. It is also suggested that further study be carried out on how to facilitate reflection in nursing education from the perspectives of instructors, physicians, and nursing professionals.

**Author Contributions:** Conceptualization, Y.A., B.A., L.B., F.A., M.S.I., A.T. and M.F.; methodology, Y.A., A.T., M.S.I. and M.F.; validation, Y.A. and M.F.; formal analysis, Y.A., B.A., L.B. and M.F; investigation, F.A., M.S.I. and A.T.; resources, Y.A.; data curation, Y.A., B.A., L.B. and M.F; writing—original draft preparation, Y.A. and M.F.; writing—review and editing, M.S.I. and M.F.; supervision, Y.A. and M.F.; project administration, M.F. All authors have read and agreed to the published version of the manuscript.

**Funding:** This research received no external funding.

**Institutional Review Board Statement:** The study was conducted in accordance with the Declaration of Helsinki and approved by the Institutional Review Board (ECO-R-103).

**Informed Consent Statement:** Informed consent was obtained from all subjects involved in the study.

**Data Availability Statement:** The data will be shared by the authors of this research paper upon request.

**Acknowledgments:** The authors want to acknowledge the efforts of the research assistant who helped in the data collection and publishing of this paper as well as the participants who were involved in the study.

**Conflicts of Interest:** The authors declare no conflict of interest.

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
