# Peer review of "Perspectives of Nursing Students on Promoting Reflection in the Clinical Setting: A Qualitative Study"

_nursrep, doi:10.3390/nursrep12030053_

Round 1

Reviewer 1 Report

Thank you for this article on reflection among nursing students. This is important research, but there are some weaknesses that need to be addressed.

1. Aim: There are inconsistencies in the study aim, and some of the stated aims are outside the remit of this study. For instance, in the abstract, the study aimed at "exploring nursing students' perspectives on the facilitators of reflective practices in the clinical setting". Then in the background, page 2 (line 85-86), the aim was stated as to "... investigate nursing students' perceptions of the reflection outcomes in the clinical context in the light of social, cultural, and educational variations". Then from lines 90-91 (page 2), it was stated as "to gain a better understanding of nursing students' opinions of the importance of reflection in clinical practice and the facilitators of such reflection. Both aims stated in page 2 are not covered in this study and this means the study did not meet its aim. To revise, the aim stated in the abstract should be the one guiding this study

2. Under the research design (page 2), the statement that states "the purpose of this study was to employ a phenomenological explorative analytical methodology" is incorrect because the aim of the study is not to employ a methodology.

3. There are many uncited assertions and assumptions in the background. For example, on page 2 (lines 54-57)

4. On page 2 (lines 69-71), the sentence claimed that there are several researches, but only one study was cited.

5. While this study was a qualitative study, many of the languages used are not qualitative in nature and need revision. For example, page 3 (line 96), the participants were called 'subjects'. Response rate of 72.41 (page 3, line 102) is also what one would expect in a quantitative study. Likewise, rather than use 'eliminated' (page 3, line 112), use 'excluded'. Instead of reliability (page 4, lines 183-184), qualitative research uses credibility. Qualitative research generates hypotheses, it doesn't test hypotheses. However, it was claimed that your data analysis "allowed the hypothesis developed to be investigated". Again this is confusing. On page 7 (line 331), kindly change or remove the word 'surveyed'. Language matters.

6. Purposive sampling: The explanation provided for purposive sampling is insufficient. In a qualitative study, the aim is not to recruit everyone in the population. Rather, it is to carefully select those who can provide rich information. But in this study, an attempt was made to recruit everyone and this undercut the claim of saturation that was made earlier on. 

7. Page 3 (lines 115-116), it was said that the reflection form was filled by students, instructor and classmates. However, the results presented were only from the students. This is confusing. Likewise, what role did the portfolio play? This did not reflect in the analysis and findings

8. Analysis: Page 4 (line 160), why were the audio tapes played repeatedly if the transcripts were transcribed accurately. This is confusing. It is important to mention the method of qualitative content analysis followed and why qualitative content analysis was the preferred method. Also, in qualitative content analysis, 'categories' are used instead of 'themes'. This disparity should be addressed.

9. Discussion: The discussion does not totally reflect the data. For example, it was claimed on page 7 (lines 333-334) that educational level is a driver of reflection. But there was no evidence of this in the data. Likewise, the discussion was a summary of other studies. It does not raise any intellectual curiosity about the importance of reflection in nursing education

10. The manuscript needs to be extensively edited for grammar, spelling and elimination of colloquial writing

Reviewer 2 Report

   Revision

Introduction.

In the introduction it is necessary to incorporate more studies that confirm the affirmations made by the authors. For example, “Students also find the procedure of reflection difficult and feel that it does not occur spontaneously, therefore it needs a safe and secure atmosphere in which they may cultivate reflection with the assistance of professionals”, on which study is this based? Some statements are debatable as they cannot be found in the evidence that the authors base their writing on; According to evidence, nursing students do not always offer the needed care for patients at the hospital, indicating a theory-practice mismatch?.

Thus, I propose to increase the review of the literature and increase the number of studies to substantiate the claims.

Goal

The objective that appears in the abstract: “This study aims at exploring nursing students' perspectives on the facilitators of reflective practices in the clinical setting” does not coincide with the one that appears at the end of the introduction: “The current study was undertaken to qualitatively investigate nursing students' perceptions of the reflection outcomes in the clinical context in the light of social, cultural, and educational variations in Saudi Arabia”

This aspect should be clarified as it is key.

Design

Please clarify whether the study is phenomenological or constructivist.

Participants

Describe how the students were recruited, by whom, and whether their participation had any implications for the assessment.

 Portfolio

This section is very theoretical; it does not describe how the student should build it and what was part of it. Nor is it understood that it has a specific section since I understand that the instrument used for data collection was the interview.  

Interview The protocol of the interview should be more detailed and it must be aligned with the objective of the study.  

Analysis of data The type of analysis performed and what part of it the authors base the described procedure on must be indicated.  

Results More details are needed about the sample; for example age.  It is very good that they are from different courses, but is the level of reflection the same in the first as in the third? They would still have to include this aspect in the limitations. The presentation of the results is correct but all the statements must be accompanied by a unit of meaning, putting them at the end of the sections confuses the reader. The format needs to be improved.   By not knowing the reference author of the analysis, themes and sub-themes are not understood, would they not be categories? Discussion The authors offer a discussion that is not always aligned with the results obtained, can it really be said that students improve the level of knowledge and skills? I believe that what the results of the study show is the need and value of professional values ​​such as self-responsibility to learn or to care for patients or ethical values ​​plus the motivation that reflection entails. It would be good if the authors highlighted these elements in the discussion.  

On the other hand, there would be the external drivers of reflection, these are better discussed in the section.      

Reviewer 3 Report

Dear authors! Thank you for this great work. In my opinion, this manuscript is interesting. However, there are some points to be considered before publication. I thing that the article requires some English language editing since numerous errors have been noted e.g. line 27 namely instead named; 39 experiencing instead experience; 42 in vice over the past years; 115 shall be filled v. should be field.....

Also, a small part of the introduction could be written more clearly lines 50-61. Especially part in lines 75-79 I think that it can even be thrown out or brought into connection with the topic of the work.

The first paragraph in the results may be moved to Setting and Sample.

I do not agree with highlighting the first limitation. The results of a qualitative study cannot even be generalized. That is not even expected. Rather, state the limitations related to the choice of sample or respondents.

Please, bring the conclusions into a clearer connection with the results.

Round 2

Reviewer 1 Report

Thank you for taking the time to thoroughly address the comments.

Some minor things to consider:

1. Colliazi's citation was not found in the reference list

2. On page 8 (lines 324 to 325), edit the statement to "According to the findings, the drivers of reflection ranged from personal and extrinsic levels".

3. There is still a need for editing. Good language editing will strengthen the use of this important work

Reviewer 2 Report

Thank you very much for being so rigorous in the modifications.

I suggest a small change but it is very important. In section 2.1 do not use the word analytical, it sounds very bad in qualitative studies: “This study employed a phenomenological explorative analytical methodology…”.

Congratulations on the job.
